

# "Spectrally gapped" random walks on networks: a Mean First Passage Time formula

Silvia Bartolucci[1], Fabio Caccioli[1,2,3], Francesco Caravelli[4] and Pierpaolo Vivo[5⋆]

**1** Dept. of Computer Science, University College London,
66-72 Gower Street, WC1E 6EA London (UK)
**2** Systemic Risk Centre, London School of Economics
and Political Sciences, WC2A 2AE, London (UK)
**3** London Mathematical Laboratory, 8 Margravine Gardens, London WC 8RH (UK)
**4** T-Division (T-4), Los Alamos National Laboratory, Los Alamos NM 87545 (USA)
**5** King's College London, Department of Mathematics, Strand, WC2R 2LS London (UK)

⋆ pierpaolo.vivo@kcl.ac.uk

## Abstract

We derive an approximate but explicit formula for the Mean First Passage Time of a random walker between a source and a target node of a directed and weighted network. The formula does not require any matrix inversion, and it takes as only input the transition probabilities into the target node. It is derived from the calculation of the average resolvent of a deformed ensemble of random sub-stochastic matrices $H = \langle H \rangle + \delta H$, with $\langle H \rangle$ rank-1 and non-negative. The accuracy of the formula depends on the spectral gap of the reduced transition matrix, and it is tested numerically on several instances of (weighted) networks away from the high sparsity regime, with an excellent agreement.



# 1  Introduction

The exploration of a complex network by a walker that hops randomly from one node to another according to a given probabilistic rule has received much attention in recent years [1–20], with many applications (see [21] for an excellent review), including the self-organization and generation of networks [22–24].

Among the most significant observables that can be studied analytically, the Mean First Passage Time (MFPT) plays a pivotal role. The MFPT is the average over many realizations of the walk of the "first-passage time" (or "first-hitting time") – defined as the number of steps taken by the walker to reach a target node from a given source node for the first time. Applications range from biology [25,26] to finance [27], ecology [28], kinetic network models [29] and many other fields (see [3,30] for reviews on first-passage problems on networks and other media). More recently, the idea that the most "important" nodes should also be those that are most rapidly reachable by others has been used to rank constituents [31–33] and to assess heterogeneity and correlations [34] in complex networks.

The computation of the MFPT involves a cumbersome inversion of a reduced matrix of transition probabilities. For this reason, any analytical treatment of the MFPT has in general proven difficult, with a number of attempts made to derive exact expressions – often valid when transition matrices have special symmetries – as well as approximate and mean field results (see Sec. 1.1 for details). In particular, unveiling the connection between the structural properties of the underlying network and the MFPT – as well as its scaling properties – is a non-trivial task for the majority of network topologies [35].

In this paper, we address these issues by proposing an approximate but explicit formula for the MFPT of a walker on directed and weighted networks. Our formula does not require matrix inversions, it depends only on the *local* information about the target node, and sheds light on the interplay between structural and spectral properties of the underlying network.

The plan of the paper is as follows. In Section 1.1 we provide the main definitions and an overview of closely related literature, while in 1.2 we announce our main result. In section 2, we reproduce for completeness the main steps leading to the main formula (5) starting from a random matrix formulation of the problem, already outlined in [36,37]. In section 3, we test our formula on different types of spectrally gapped (weighted) networks. Finally, in section 4 we offer some concluding remarks and outlook for future researches. The two Appendices are devoted to technical calculations and examples.

## 1.1  Setting and related works

Consider a weighted strongly connected network with $N$ nodes described by the (real valued, and not necessarily symmetric) adjacency matrix $A$. Given a source node $i$ and a different target node $j$, the MFPT $m_{ij}$ satisfies the following recurrence equation [1,38–40]

$$m_{ij} = 1 + \sum_{\ell \neq j}^{N} T_{i\ell} m_{\ell j} \,, \tag{1}$$

where the matrix element $T_{i\ell}$

$$T_{i\ell} = \frac{A_{i\ell}}{\sum_r A_{ir}} \tag{2}$$

encodes the transition probability of the walker from node $i$ to node $\ell$ (with $\sum_\ell T_{i\ell} = 1$ for all $i = 1, \ldots, N$ by normalization). The meaning of eq. (1) is straightforward: in its first step, the walker hops from node $i$ to node $\ell$, which produces the $+1$ on the right-hand side. Then, if the target has not been reached, we have to assign a weight to the MFPT from the "new" source node $\ell$ to the target $j$, which is the probability of reaching the "new" source node $\ell$ from the "old" source node $i$. This produces the second term on the right-hand side.

There are different strategies to extract meaningful information from (1). On the one hand, one could simply iterate the equation numerically – given the network instance and the diffusion protocol, encoded in the matrix $T$ – until convergence is reached [40]. Alternatively, for a given target node $j$, one could rewrite the equation in the vector-matrix form

$$\mathbf{m}^{(j)} = \mathbf{1} + T^{(j)}\mathbf{m}^{(j)} \Rightarrow \mathbf{m}^{(j)} = (\mathbb{1} - T^{(j)})^{-1}\mathbf{1} \,, \tag{3}$$

where all quantities are $(N-1)$-dimensional: $\mathbf{1}$ is the column vector of ones, $\mathbb{1}$ is the identity matrix, and $T^{(j)}$ is the transition matrix where the $j$-th row and column have been erased [41]. The resulting vector $\mathbf{m}^{(j)}$ encodes all MFPT to the target node $j$ starting from all other nodes in the network. The seemingly harmless eq. (3) has however a few important drawbacks: (i) it requires the inversion of a possibly large and ill-conditioned matrix, which makes a numerical approach prone to inaccuracies [42], (ii) the nonlinear relation between $\mathbf{m}^{(j)}$ and $T^{(j)}$ makes it difficult to infer the functional dependence of the former on network parameters (e.g. the mean degree) from the knowledge of the latter – unless the transition matrix enjoys special symmetries or internal structure, and (iii) it implicitly takes for granted that the *full* adjacency matrix of the underlying network is known with great accuracy, which may not necessarily be the case in practical applications.

Another exact approach – pioneered by Noh and Rieger [43] – relies on the identity $m_{ij} = \sum_{n \geq 0} nF_{ij}(n)$, where $F_{ij}(n)$ is the probability that the walker starting from $i$ arrives in $j$ for the first time after $n$ moves. Using the Markov property of the walk, and suitable generating functions (see [21] for details) it is possible to write an expression for $m_{ij}$ in terms of the series coefficients of the (discrete) Laplace transform of $p_{ij}(n)$ – the probability that the walker starting in $i$ reaches $j$ after $n$ moves. Although exact, the final formula can be opaque to interpretations, unless the transition matrix has again special symmetries or structure that make the master equation analytically tractable [44]. These cases are a rare luxury, though.

Finally, there are a number of approximate results, using e.g. a mean-field approach [45–48]. The crudest approximation consists in noticing that – regardless of the source node – the target node $j$ is reached with an approximate probability of $p_j^\star$ in each time step, where $\mathbf{p}^\star$ is the equilibrium probability vector of the Markov transition matrix. Therefore,

$$m_{ij} \approx \sum_{k=1}^{\infty} kp_j^\star(1-p_j^\star)^{k-1} = \frac{1}{p_j^\star}\,. \tag{4}$$

The estimate in (4) can be rather loose, and $m_{ij}$ may deviate considerably from $1/p_j^\star$. More sophisticated mean-field approaches have been devised, which perform better in certain situations [45, 49, 50]. For discussion of scaling theory based on renormalization theory for first-passage time and other quantities on networks, see [51, 52]. For analytical approaches to MFPT based on spectral theory and generating functions, see [17, 53–57]. For other approaches and applications of first-passage times and return times on networks, see [58–62].

## 1.2 Summary of main result

In this paper, we put forward a novel approximate formula for $m_{ij}$, which we shall show in the following to be

$$m_{ij} \approx 1 + (N-1)\frac{1 - T_{ij}}{\sum_{\ell \neq j} T_{\ell j}} \ . \tag{5}$$

Our formula is valid on a generic (directed, weighted, strongly connected) network, provided that its reduced transition matrix $T^{(j)}$ – obtained by removing the $j$-th row and column – has a "large" spectral gap, defined as $\lambda_1 - \max\{|\lambda_2|, \ldots, |\lambda_{N-1}|\}$ with $\lambda_1 \in (0,1)$ the Perron-Frobenius eigenvalue, and the $\{\lambda_i\}$ being the other eigenvalues of $T^{(j)}$ in the complex plane. Since the spectral gap tends to diminish the sparser the network becomes [63–65], the formula (5) is not suitable for "too sparse" networks.

The formula (5) is strikingly simple, and – in spite of being obtained in a large-$N$ setting – we find that it is very accurate also for spectrally gapped walks on relatively small networks, as we demonstrate below. It is obtained by approximating the reduced transition matrix $T^{(j)}$ as a rank-1, sub-stochastic matrix: from each node $i \neq j$, the walker may either hop on $j$ directly (with probability $T_{ij}$), or hop on any of the other $N-1$ nodes – connected to $i$, or not – with the same probability $(1 - T_{ij})/(N-1)$. Within this approximation, only the neighborhood of the target node really matters – which reveals an interesting approximate symmetry: two sufficiently dense networks that share the same set of transition probabilities into a given node $j$, also share the full set of MFPTs from any source node into the target $j$, irrespective of how "unlike" each other they are away from $j$. For simple diffusion on a fully connected network (where $T_{ij} = 1/(N-1)$ for all $i \neq j$), our formula (5) reduces to the known (exact) result $m_{ij} = N - 1$ – which holds true also for sufficiently dense Erdős-Rényi networks [66, 67], independently of the probability $p \sim \mathcal{O}(1)$ that each node pair has an edge between them. For a detailed discussion of results on MFPT on different kinds of graphs and fractal structures, see again [21] and references therein.

## 2 Sketch of the proof

In this section, we report for completeness the random matrix calculation that we outlined elsewhere [36, 37] in another context, from which the approximate formula (5) follows as an immediate corollary.

The main idea is to replace a given (empirical) reduced transition matrix $T^{(j)}$ with a random sub-stochastic matrix (called $H = (h_{\ell m})$ in the following), in such a way that "some" macroscopic features of $T^{(j)}$ are retained in $H$. More specifically, our model assumes that the row sums are preserved (on the average), i.e. $z_i := \sum_k T_{ik}^{(j)} = \sum_k \langle h_{ik} \rangle$ – but these sums are spread "as evenly as possible" among the columns of $H$.

Consider therefore a random $N \times N$ matrix $H = (h_{\ell m})$ with $h_{\ell m} \geq 0$, which can be written as

$$H = \langle H \rangle + \delta H \ . \tag{6}$$

The deterministic rank-1 matrix $\langle H \rangle$ reads

$$\langle H \rangle = \begin{pmatrix} \frac{z_1}{N} & \cdots & \frac{z_1}{N} \\ \vdots & \ddots & \vdots \\ \frac{z_N}{N} & \cdots & \frac{z_N}{N} \end{pmatrix}, \tag{7}$$

in terms of positive constants $\{z_1, \ldots, z_N\}$. With this definition, the random matrix $H$ is essentially a "noise-dressed" version of the rank-1 (balanced) matrix $\langle H \rangle$, whose row sums are

$\{z_1, \ldots, z_N\}$. The entries – not necessarily independent – of the random perturbation $\delta H$ satisfy $\langle \delta h_{\ell m} \rangle = 0$ for all $\ell, m$. The average $\langle \cdot \rangle$ is taken w.r.t. the joint probability density of the entries of the matrix $\delta H$.

Clearly, $\langle H \rangle$ has a single non-zero, real and positive eigenvalue $\lambda_1 = \frac{1}{N} \sum_\ell z_\ell \equiv \bar{z}$, and $N - 1$ zero eigenvalues – and therefore a spectral gap of $\sim \mathcal{O}(1)$. Assume that the spectral radius[1] of $H$ satisfies $\rho(H) < 1$.

If $\delta H$ were identically zero, the vector[2]

$$\mathbf{m} = (\mathbb{1} - H)^{-1} \mathbf{1} \tag{8}$$

could be computed exactly using the Sherman-Morrison formula [69] to give

$$m_\ell = 1 + \frac{z_\ell}{1 - \bar{z}} \, , \tag{9}$$

with $\bar{z} = (1/N) \sum_{i=1}^N z_i$.

In presence of a random perturbation $\delta H$, we ask what the *average* value of $\mathbf{m}$ would be,

$$\langle \mathbf{m} \rangle = \langle (\mathbb{1} - H)^{-1} \mathbf{1} \rangle \, , \tag{10}$$

and in particular how small should the perturbation $\delta H$ be to ensure that the Sherman-Morrison result (9) keeps holding on average – to leading order in $N$ – even in this "noise-dressed" case. It turns out that $\delta H$ must satisfy a certain cumulant decay law (see eq. (21) below). The condition on the spectral radius $\rho(H) < 1$ ensures instead that the inverse matrix on the r.h.s. of (10) exists.

It is instructive to see what happens in the special case of an i.i.d. Gaussian perturbation with $\langle \delta h_{ij}^2 \rangle = \sigma_N^2$ for all $i, j$, for which fuller analytical considerations are possible. Let us reverse momentarily the roles of $\delta H$ and $\langle H \rangle$. We would essentially have here a real Gaussian and non-symmetric matrix $\delta H$ (hence belonging to the Ginibre ensemble [70]), which is deformed by a rank-1 matrix $\langle H \rangle$. This problem – albeit with the additional twist that we require positivity of the final matrix – is relatively well-understood in Random Matrix Theory [71–74]. In the absence of the rank-1 deformation, the spectrum of $\delta H$ would fill a circle in the complex plane with radius $r_N = \sqrt{N} \sigma_N$ – this is known as Girko-Ginibre circular law. However, the addition of the rank-1 deformation $\langle H \rangle$ leaves the circular bulk of eigenvalues unperturbed, but may lead to the appearance of an extra isolated outlier at $\lambda_{out} = \bar{z} \sim \mathcal{O}(1)$. Choosing "too big" a variance $\sigma_N^2$ has therefore two harmful effects: (i) positivity of the matrix entries of $H$ is no longer guaranteed,[3] and (ii) all the eigenvalues of $H$ become of the same order, with the circular bulk swallowing up the outlier and annihilating the spectral gap. A similar clash between the positivity constraint (leading to a Perron-Frobenius outlier) and the standard circular law for Gaussian matrices – leading to a phase transition – was recently noted in [75].

In order to get a large spectral gap, we need to require[4]

$$r_N = o(1) \Rightarrow \sigma_N = o(1/\sqrt{N}) \, . \tag{11}$$

In Fig. 1 we plot on the left the typical spectrum of a randomly generated matrix of the form (6), with i.i.d. Gaussian $\delta h_{ij}$ having[5] $\sigma_N \sim \mathcal{O}(1/N)$, while on the right we have

---

[1] The spectral radius is $\rho(H) = \max_i |\lambda_i|$. For simplicity, we will call *sub-stochastic* a matrix $H$ satisfying $\rho(H) < 1$ $(\star)$, instead of $\sum_\ell h_{i\ell} < 1$ for all $i$ $(\star\star)$, even though the implication is only in one direction, $(\star\star) \Rightarrow (\star)$ [68].

[2] We use the notation $\mathbf{m}$ to keep contact with the MFPT vector defined in Eq. (3). The connection between the two objects will become clear very shortly.

[3] By this, we mean that the probability of drawing a negative entry would not be exponentially small.

[4] We use the little-$o$ notation indicating that $f_N = o(g_N)$ if $\lim_{N \to \infty} f_N / g_N = 0$.

[5] We intentionally choose a smaller variance than strictly needed in (11) to make the gap as visible as possible in Fig. 1.

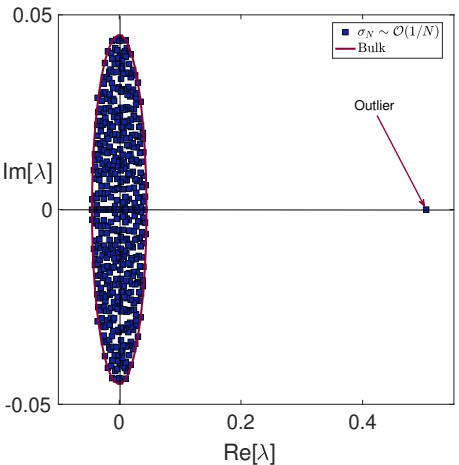
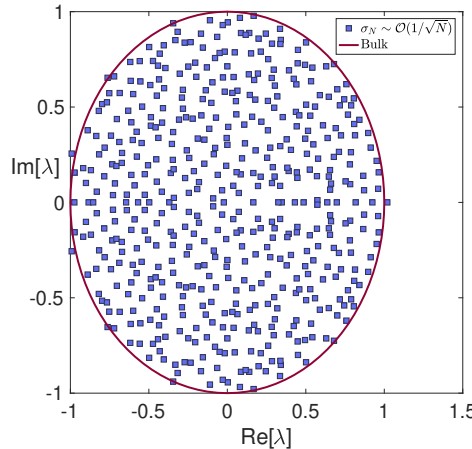

Figure 1: **Left:** Spectrum of a typical instance of a $N = 500$ Gaussian matrix (with $\sigma_N \sim \mathcal{O}(1/N)$) plus a rank-1 deformation of the form (7). The red line encloses the circle of radius $r_N \sim \mathcal{O}(1/\sqrt{N})$, while an isolated outlier at $\lambda_{out} = \bar{z}$ is clearly visible. **Right:** Spectrum of a typical instance of a $N = 500$ Gaussian matrix (with $\sigma_N \sim \mathcal{O}(1/\sqrt{N})$) plus a rank-1 deformation of the form (7). The red line encloses the circle of radius $r_N \sim \mathcal{O}(1)$, which swallows the would-be outlier at $\lambda_{out} = \bar{z}$ altogether. The same constant values $\{z_1, \dots, z_N\}$ have been used to produce the two plots.

$\sigma_N \sim \mathcal{O}(1/\sqrt{N})$. One clearly observes that the relative fluctuation $\sigma_N / \langle H \rangle_{ij}$ is too large in the latter case to guarantee positivity and a large enough gap.[6]

This simple numerical experiment – consistent with the analytical estimate in (11) – shows that a *generic* positive and sub-stochastic matrix (provided its spectral gap is "large") can be interpreted as the superposition of a rank-1 matrix (fully determined by the original row sums) and a Gaussian noise matrix with sufficiently small variance. We will show in section 3 that such large-gap matrices appear naturally in the treatment of MFPT on weighted networks away from the high sparsity regime.

To compute (10), one first defines the $2N \times 2N$ Hermitian matrix

$$B(\eta) = \begin{pmatrix} -\mathrm{i}\eta\mathbb{1} & \mathbb{1} - H^T \\ \mathbb{1} - H & -\mathrm{i}\eta\mathbb{1} \end{pmatrix}, \tag{12}$$

where i is the imaginary unit, and $\eta$ is a small regularizer that ensures that $B^{-1}$ exists.

Using the formula for the inverse of a block matrix, it is possible to show [37] that

$$\langle m_\ell \rangle = \lim_{\eta \to 0} \sum_{k=N+1}^{2N} \left\langle [B^{-1}(\eta)]_{\ell,k} \right\rangle, \qquad \ell = 1, \dots, N. \tag{13}$$

Next, we use the following result: given a (complex) symmetric matrix $M$ of size $N \times N$, with purely imaginary diagonal elements $M_{ii} = -\mathrm{i}m_{ii}$, with $m_{ii} > 0$, the following formula holds

$$[M^{-1}]_{ab} = \mathrm{i}\frac{\int d\mathbf{x}\, x_a x_b \exp\left[-\frac{\mathrm{i}}{2} \sum_{i,j}^N x_i M_{ij} x_j\right]}{\int d\mathbf{x}\, \exp\left[-\frac{\mathrm{i}}{2} \sum_{i,j}^N x_i M_{ij} x_j\right]}, \tag{14}$$

---

[6]Such a large $\sigma_N$ would of course violate the cumulant decay condition (21) in the special case of Gaussian i.i.d. $\delta h_{ij}$, which again requires $\sigma_N = o(1/\sqrt{N})$ (see Appendix A for details).

where $\mathbf{x}$ denotes a $N$-dimensional vector, and the integrals run over $\mathbb{R}^N$ [76].

Applying this formula to the $2N \times 2N$ matrix $B(\eta)$, and inserting it in Eq. (13), we get the following integral representation of the $\ell$-th element of the vector $\mathbf{m}$ in (10)

$$\langle m_\ell \rangle = -\mathrm{i} \lim_{\eta \to 0} \lim_{\omega, \xi \to \mathbf{0}} \left\langle \frac{Z_1(\omega, \xi, H)}{Z(H)} \right\rangle, \tag{15}$$

where

$$Z_1(\omega, \xi, H) = \sum_{m=1}^{N} \partial_{\omega_\ell} \partial_{\xi_m} \int \mathrm{d}\mathbf{x} \mathrm{d}\mathbf{y} \, \exp\left[ -\frac{\eta}{2} \sum_{i=1}^{N} (x_i^2 + y_i^2) - \mathrm{i} \sum_{i=1}^{N} x_i y_i + \mathrm{i} \sum_{i=1}^{N} \omega_i x_i \right.$$
$$\left. + \mathrm{i} \sum_{i=1}^{N} \xi_i y_i + \mathrm{i} \sum_{i,j=1}^{N} x_i h_{ji} y_j \right],$$

$$Z(H) = \int \mathrm{d}\mathbf{x} \mathrm{d}\mathbf{y} \, \exp\left[ -\frac{\eta}{2} \sum_{i=1}^{N} (x_i^2 + y_i^2) - \mathrm{i} \sum_{i=1}^{N} x_i y_i + \mathrm{i} \sum_{i,j}^{N} x_i h_{ji} y_j \right]. \tag{16}$$

Using the "replica trick" [77–79]

$$\frac{Z_1}{Z} = \lim_{n \to 0} Z_1 Z^{n-1}, \tag{17}$$

where the variable $n$ is initially promoted to an integer, we get rid of the denominator and land on

$$\langle m_\ell \rangle = -\mathrm{i} \lim_{n \to 0} \lim_{\eta \to 0} \lim_{\omega, \xi \to \mathbf{0}} \sum_{m=1}^{N} \partial_{\omega_\ell} \partial_{\xi_m} \int \prod_{a=1}^{n} \mathrm{d}\mathbf{x}_a \mathrm{d}\mathbf{y}_a \times \tag{18}$$

$$\exp\left[ -\frac{\eta}{2} \sum_{i=1}^{N} \sum_{a=1}^{n} (x_{ia}^2 + y_{ia}^2) - \mathrm{i} \sum_{i,a} x_{ia} y_{ia} + \mathrm{i} \sum_{i=1}^{N} \omega_i x_{i1} + \mathrm{i} \sum_{i=1}^{N} \xi_i y_{i1} \right] \Phi(\{\mathbf{x}_a\}, \{\mathbf{y}_a\}),$$

where

$$\Phi(\{\mathbf{x}_a\}, \{\mathbf{y}_a\}) = \exp\left( \frac{\mathrm{i}}{N} \sum_{i,j}^{N} z_j \phi_{ij} \right) \varphi(\{\phi_{ij}\}), \tag{19}$$

with

$$\varphi(\{\theta_{ij}\}) = \left\langle \mathrm{e}^{\mathrm{i} \sum_{i,j}^{N} \delta h_{ji} \theta_{ij}} \right\rangle \tag{20}$$

being the joint cumulant generating function of the entries of the matrix $\delta H$, and $\phi_{ij} = \sum_a x_{ia} y_{ja} \sim \mathcal{O}(1)$. Assuming the following cumulant decay condition

$$\log \varphi(\{\phi_{ij}\}) = o(N), \tag{21}$$

for large $N$, we can neglect all higher-order terms and land on a "replicated" version of the Sherman-Morrison formula for the matrix $\langle H \rangle$ alone, yielding eventually[7]

$$\langle m_\ell \rangle = 1 + \frac{z_\ell}{1 - \bar{z}} + o(1), \tag{22}$$

---

[7]In the Gaussian case, the $o(1)$ correction terms can be estimated more accurately as $\mathcal{O}(1/N^2)$, further confirming that the leading order term is already an excellent approximation for even a moderately small $N$.

with $\bar{z} = (1/N) \sum_{i=1}^{N} z_i$.

In summary, provided that the cumulants of $\delta h_{ij}$ decay sufficiently fast for large $N$, the "noise-dressing" of the average, rank-1 matrix $\langle H \rangle$ is inconsequential, and the Sherman-Morrison formula is universal (noise-independent) to leading order in $N$. Although for the most general, arbitrarily correlated noise term $\delta H$ it is difficult to establish formally that the cumulant decay condition (21) is equivalent to $H$ having a "large" spectral gap, the intuition garnered from the Gaussian case leads us to conjecture this must be generally the case.

The formula (5) for the MFPT readily follows from the identification $H \equiv T^{(j)}$, and $z_\ell \equiv 1 - T_{\ell j}$. This is due to $z_\ell$ being the (average) sum of the $\ell$-th row of $H \equiv T^{(j)}$, and $T^{(j)}$ being obtained by the row-stochastic matrix $T$ ($\sum_k T_{\ell k} = 1$) by erasing the $j$-th row and column.

## 3 Network examples

In this section, we apply our formula to walks on different network instances (fully connected, Erdős-Rényi, random regular). More precisely, we now test Eq. (5) against (i) exact evaluations of formula (3) for the MFPT via direct matrix inversion, and (ii) numerical simulations of random walks. We do not report here on (dense) scale-free topologies, whose phenomenology is very similar to the other cases, albeit with significantly larger fluctuations: a detailed study of this (and other) heterogeneous cases is deferred to a separate publication. We do, however, test on a simple and exactly solvable case (the *star graph* with $N$ nodes) the hypothesis that the accuracy of our approximate formula (5) may vary (within the *same* instance) from node to node, depending on how well connected the target node is to the rest of the network (see Appendix B for details).

### 3.1 Fully Connected

We consider a single instance of a fully connected, directed, weighted network with $N = 500$. Each link is endowed with a random weight sampled from a uniform distribution in $[0, 1]$. In all cases below, we have checked that nothing changes with other edge weights (e.g. exponential). We fix a source node $i$ and a target node $j$, and we consider random walks starting in $i$ and hitting $j$ for the first time after $m_{ij}$ hops, performed according to the transition probabilities in Eq. (2). In Fig. 2 we see (i) a plot of the eigenvalue spectrum in the complex plane for a typical instance of the sub-stochastic matrix $T^{(j)}$, obtained erasing the $j$-th row and column from the original transition matrix $T$ defined in eq. (2), and (ii) a scatter plot of exact MFPT (eq. (3)) vs. our approximate formula (5) – each point in the plot refers to a randomly picked $(i, j)$ pair on a randomly generated instance of the graph. We observe that the points nicely follow the straight line with slope 1.

To the best of our knowledge, at present there are no exact and explicit formulas available for the MFPT on a *weighted* and *directed* fully connected network, in spite of the very simple geometry of the system. Our approximate formula does an excellent job while requiring as input only the $N-1$ incoming weights into the target node $j$ – all the other information away from $j$ being entirely irrelevant. Indeed, given one instance $T$ of the transition matrix, we have constructed another synthetic instance $T'$ such that – for a prescribed node $j$ – we have $T_{ij} = T'_{ij}$ for all $i$. All the other entries in each row of $T'$ are randomly reshuffled with respect to the corresponding row of $T$, to preserve the row sum constraint $\sum_\ell T_{i\ell} = \sum_\ell T'_{i\ell} = 1$. We have checked that the two walkers $T$ and $T'$ share the full set of MFPTs into node $j$, as expected.

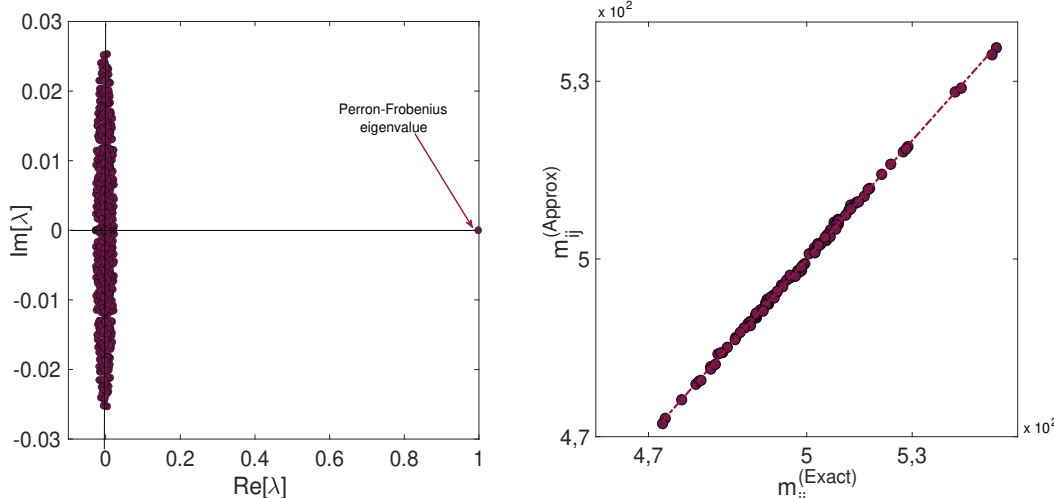

Figure 2: **Left:** eigenvalues in the complex plane for a typical instance of $T^{(j)}$ for the fully connected network described in section 3.1 with edge weights drawn from a uniform $[0, 1]$ distribution. **Right:** scatter plot of exact results (obtained by matrix inversion, see eq. (3)) vs. our approximate formula in eq. (5). Each point (100 in total) refers to a randomly picked $(i, j)$ pair on a randomly generated instance of the graph. In dashed red, the straight line with slope $= 1$, indicating perfect agreement.

## 3.2 Erdős-Rényi

We now consider a single instance of a directed Erdős-Rényi (ER) network with $N = 500$. Nodes are randomly connected with probability $p = c/N$, for different values of the mean connectivity $c$. The elements of the weighted adjacency matrix are given by $\tilde{A}_{ij} = C_{ij}K_{ij}$, where the symmetric $\{0, 1\}$ matrix $C$ includes information about the connectivity of the graph, whereas the $K_{ij}$ are independently sampled from a uniform distribution in $[0, 1]$, without any symmetry constraint. Next, we isolate the strongly connected component of $N_{sc} \leq N$ nodes[8] (using a depth first search algorithm), and denote by $A$ the restriction of $\tilde{A}$ to the nodes in the strongly connected component. We fix a source node $i$ and a target node $j$ in the strongly connected component, and we consider random walks starting at $i$ and hitting $j$ for the first time after $m_{ij}$ hops, performed according to the transition probabilities in eq. (2). In Fig. 3 we see (i) for two different values of $c$, plots of the eigenvalue spectrum in the complex plane of a typical instance of the sub-stochastic matrix $T^{(j)}$, obtained erasing the $j$-th row and column from the original transition matrix $T$ defined in eq. (2), and (ii) a scatter plot of exact MFPT (eq. (3)) vs. our approximate formula (5) – each point in the plot refers to a randomly picked $(i, j)$ pair on a randomly generated instance of the graph. We observe that (i) the spectral gap decreases the smaller the connectivity $c$ becomes, and – as expected – (ii) the points nicely follow the straight line with slope 1 for higher $c$, whereas the accuracy deteriorates as the network becomes sparser.

This behavior is further corroborated qualitatively in Table 1, where we report – for a specific $(i, j)$ pair on randomly generated (single) instances of ER networks with uniform weights and initial size $N = 500$ – results from the numerical average over $M = 10000$ walks (*simulations*), the corresponding *exact* result in (3), and our *approximate* formula (5). The agreement is still within a few percent of the exact result, even for a reasonably low $c$ ($c = 10$), while it deteriorates dramatically only below the connectedness threshold $c \approx \log N = 6.21$.

---

[8]For $c > \ln N$, the graph is almost surely connected ($N_{sc} \equiv N$), while for $c < \ln N$ it almost surely contains isolated nodes ($N_{sc} < N$).

Table 1: Comparison between simulations, exact, and approximate value for the MFPT on weighted ER networks of different mean connectivity $c$.

| $c$ | Simulations | Exact (3) | Approximate (5) |
|---|---|---|---|
| 250 | 539.11 | 542.66 | 542.16 |
| 190 | 486.09 | 482.57 | 477.87 |
| 100 | 500.20 | 502.82 | 508.09 |
| 50 | 736.03 | 713.37 | 692.51 |
| 10 | 499.30 | 495.04 | 474.66 |
| 4 | 458.37 | 464.61 | 329.45 |

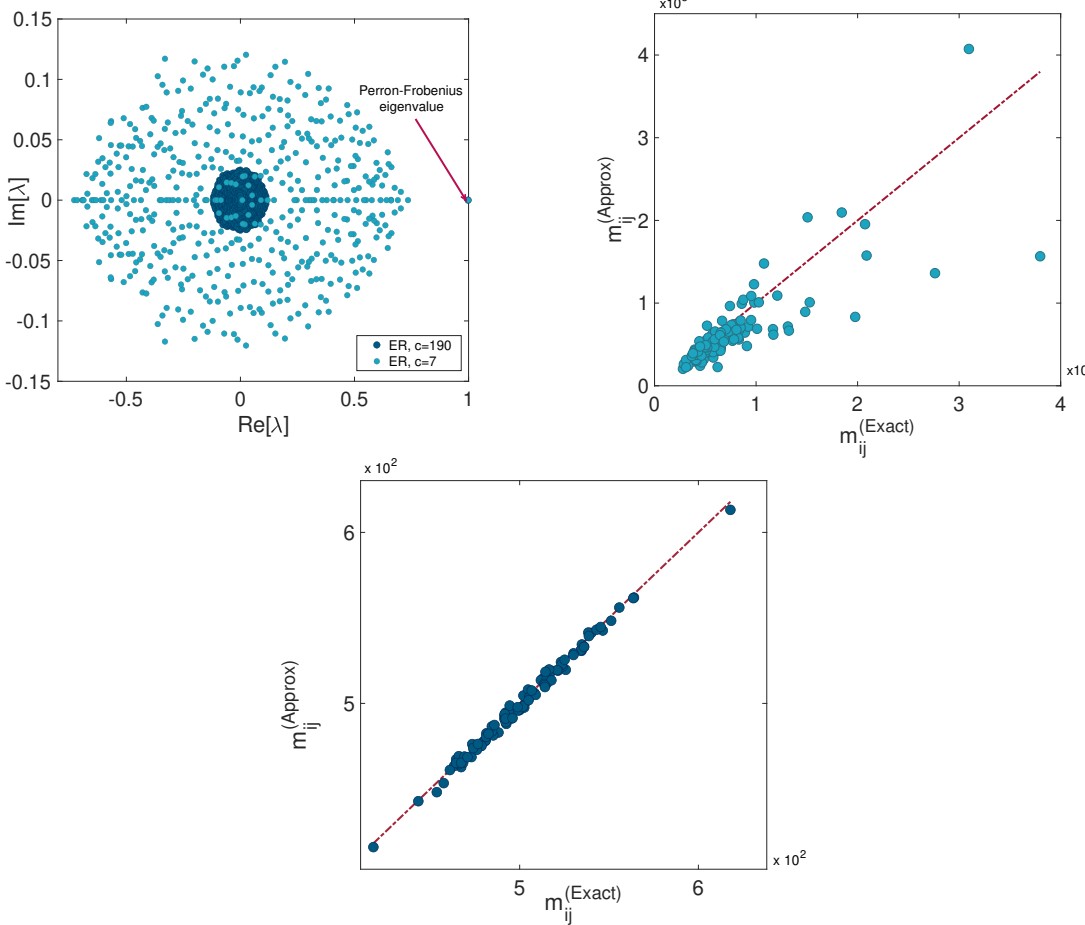

Figure 3: **Top left:** eigenvalues in the complex plane for a typical instance of $T^{(j)}$ for the Erdős-Rényi network described in section 3.2, with edge weights drawn from a uniform $[0,1]$ distribution, and two values of $c$ ($c = 190$ (dark blue dots) and $c = 7$ (light blue dots)). Clearly the spectral gap is much narrower for the low-$c$ case. **Middle:** scatter plot of exact results (obtained by matrix inversion, see eq. (3)) vs. our approximate formula in eq. (5). Each point (100 in total) refers to a randomly picked $(i, j)$ (source-target) pair, on a randomly generated instance of the graph with $N = 500$ and $c = 190$. In dashed red, the straight line with slope = 1, indicating perfect agreement. **Top right:** same scatter plot as the middle panel, but with $c = 7$ instead.

## 3.3   Random Regular

We now consider a single instance of a Random Regular Graph (RRG), with $N = 500$ nodes, constructed using the matching algorithm described in [80] and references therein. Each node has exactly $c$ neighbors, and each link is endowed with a random weight sampled from a uniform distribution in $[0, 1]$. In Fig. 4 we see (i) for $c = 190$ and $c = 7$, plots of the eigenvalue spectrum in the complex plane for a single instance of the sub-stochastic matrix $T^{(j)}$, obtained erasing the $j$-th row and column from the original transition matrix $T$ defined in eq. (2), and (ii) scatter plots of exact MFPT (eq. (3)) vs. our approximate formula (5) – each point in the plot refers to a randomly picked $(i, j)$ pair on a randomly generated instance of the graph. We observe again that (i) the spectral gap decreases the smaller the connectivity $c$ becomes [63–65], and – as expected – (ii) the points nicely collapse on the straight line with slope 1 for higher $c$, whereas the accuracy deteriorates as the network becomes sparser.

This behavior is further corroborated qualitatively in Table 2, where we report – for a specific $(i, j)$ pair on randomly generated (single) instances of RRGs with uniform weights and size $N = 500$ – results from the numerical average over $M = 10000$ walks (*simulations*), the corresponding *exact* result in (3), and our *approximate* formula (5). The agreement is still within $\sim 2\%$ percent of the exact result for $c$ as low as 50. However, the evidence provided in Tables 1 and 2 about the relative accuracies should be taken with some caution, as the reported numerical values are highly sensitive to the precise instance of the graph at hand, as well as the pair of nodes chosen.

Table 2: Comparison between simulations, exact, and approximate value for the MFPT on weighted random regular networks of different connectivity $c$.

| $c$ | Simulations | Exact (3) | Approximate (5) |
|---|---|---|---|
| 250 | 494.07 | 492.14 | 491.08 |
| 190 | 483.79 | 479.70 | 478.65 |
| 100 | 474.14 | 472.14 | 471.43 |
| 50 | 530.01 | 524.66 | 515.85 |
| 10 | 559.02 | 554.90 | 453.31 |
| 4 | 990.92 | 993.19 | 408.10 |

## 4   Conclusions

We have derived the approximate formula (5) for the Mean First Passage Time of a walker between a source node $i$ and a target node $j$ of a (weighted and directed) strongly connected network of $N$ nodes. The formula does not require any (possibly costly and inaccurate) matrix inversion, and takes it as input only the local transition weights into the target node. Its accuracy depends on the existence of a "large" spectral gap between the Perron-Frobenius eigenvalue and the blob of all other eigenvalues of the reduced (sub-stochastic) transition matrix $T^{(j)}$ – obtained from the full transition matrix of the walker by erasing the target node's row and column. We have shown that – for a variety of "not too sparse" networks – this condition is not hard to materialize, and leads to an excellent agreement of our approximate formula with numerical simulations as well as the exact formula (3). While our approach continues to work for (sufficiently dense) *heterogeneous* networks when the target node is well-connected to the rest of the graph, the intra-row fluctuations around the random matrix

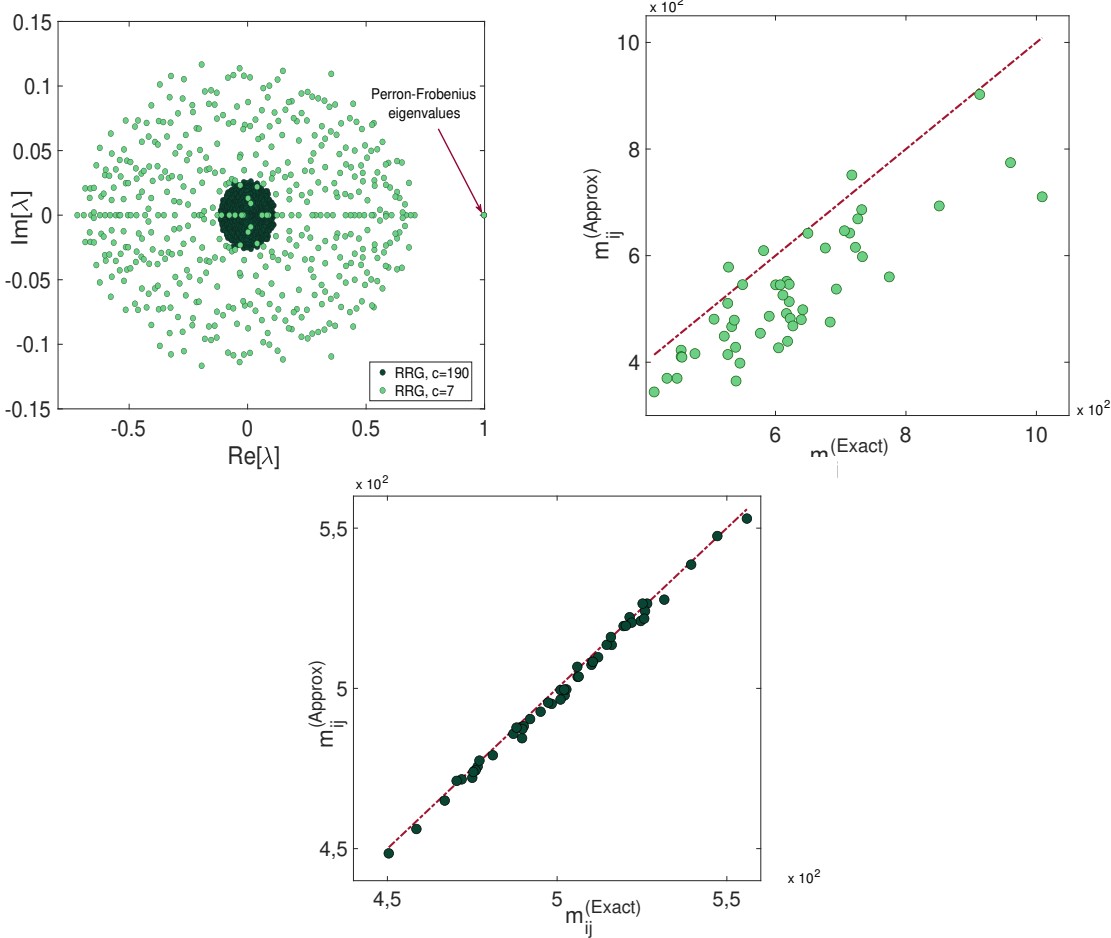

Figure 4: **Top left:** eigenvalues in the complex plane for a typical instance of $T^{(j)}$ for the Random Regular network described in section 3.3 with edge weights drawn from a uniform $[0,1]$ distribution, and two values of $c$ ($c = 190$ (dark green dots) and $c = 7$ (light green dots)). Clearly the spectral gap is much narrower for the low-$c$ case. **Middle:** scatter plot of exact results (obtained by matrix inversion, see eq. (3)) vs. our approximate formula in eq. (5). Each point (50 in total) refers to a randomly picked $(i, j)$ (source-target) pair, on a randomly generated instance of the graph with $N = 500$ and $c = 190$. In dashed red, the straight line with slope $= 1$, indicating perfect agreement. **Top right:** same scatter plot as the middle panel, but with $c = 7$ instead.

assumption $\langle h_{ij} \rangle = z_i/N$ may be very significant there and will thus require a more careful treatment.

To our knowledge, our formula (5) is one of the very few, general, and explicit results available in the literature for MFPT on *weighted* and *directed* networks (i.e. when the diffusion of the walker is biased by the edge weights), and reduces to known results for the fully connected and dense Erdős-Rényi cases when the diffusion is unbiased.

The formula would be exact if $T^{(j)}$ were a rank-1 matrix with $[T^{(j)}]_{k\ell} = (1 - T_{kj})/(N-1)$ for all $\ell$ (see eq. (9)). It is also exact to leading order in $N$ as the *average* value of the MFPT over an ensemble of random matrices with prescribed average $\langle T^{(j)} \rangle$ and sufficiently "narrow" fluctuations: this is the result of the random matrix calculation we first outlined in [36, 37],

which is reported here in section 2 for completeness.

Our work leaves a few questions open that would be very interesting to tackle in future studies:

1. How to formally prove the conjecture that a random ensemble satisfying the cumulant decay condition (21) necessarily has a large spectral gap, without assuming Gaussianity?

2. Is it possible to characterize how the accuracy of our formula (5) depends on network observables (e.g. average connectivity and other structural properties of the underlying network), which in turn influence the spectral gap?

3. In [36,37] we observed that an "improved" formula (22) could be obtained by assuming that not only the row sums of the sub-stochastic matrix $A$ were known, but also the column sums. It would be interesting to see what effect the inclusion of information about the column sums of $T^{(j)}$ might have on the final formula (5).

4. From the random matrix viewpoint, it would be very interesting to consider the model (6) with the hard constraint of positivity (which of course bounds the fluctuations of $\delta H$ and precludes Gaussianity). Alternatively, one could consider a "soft" version of the positivity constraint for a deformed Ginibre ensemble, where one bounds the probability of having negative entries and studies in more details what constraints this poses on the spectrum in the complex plane. The investigation of a "phase transition" whereby the Perron-Frobenius outlier is swallowed by the spectral bulk as the variance of $\delta H_{ij}$ increases is particularly interesting and timely (see [75]).

5. Studying more systematically how the accuracy of the main formula (5) – related to the spectral gap of $T^{(j)}$ – varies with the choice of the target node $j$ on the same instance of a heterogeneous graph is also an interesting question that deserves further investigation (see Appendix B for a preliminary attempt). Along the same lines, also the impact of degree-degree correlations on the accuracy of the formula would be interesting to study in greater detail.

These directions will be the focus of future research.

## Acknowledgements

P. Vivo is grateful to Yan V. Fyodorov and Peter Sollich for insightful discussions, to Vito A. R. Susca for advice on the numerical implementation of the RRG case, and to Yanik-Pascal Förster and A. Annibale for collaborations on a related topic. We are grateful to Naoki Masuda, Giovanni Cicuta, Eric Degiuli, Raffaella Burioni, and Luca Dall'Asta for helpful comments on a preliminary version of the manuscript.

**Funding information** The work of F. Caravelli was carried out under the auspices of the NNSA of the U.S. DoE at LANL under Contract No. DE-AC52-06NA25396, and financed via LDRD grant PRD20190195.

# A    Joint cumulant generating function for Gaussian $\delta h_{ij}$

In this Appendix, we compute explicitly the joint cumulant generating function (20) for the case of i.i.d. Gaussian entries $\delta h_{ij}$ with mean zero and variance $\sigma_N^2$. We have

$$\varphi\left(\{\theta_{ij}\}\right) = \left\langle e^{i\sum_{i,j}^N \delta h_{ji}\theta_{ij}} \right\rangle = \prod_{i,j} \int \frac{dx}{\sqrt{2\pi\sigma_N^2}} \exp\left[-\frac{x^2}{2\sigma_N^2} + ix\theta_{ij}\right]$$

$$= \exp\left[-\frac{1}{2}\sigma_N^2 \sum_{i,j}\theta_{ij}\right] . \tag{A.1}$$

Assuming $\theta_{ij} \sim \mathcal{O}(1)$, the cumulant decay condition (21) requires

$$N^2\sigma_N^2 = o(N) \Rightarrow \sigma_N = o(1/\sqrt{N}), \tag{A.2}$$

as stated earlier.

# B    The accuracy of (5) and heterogeneous networks: the star graph case

In this Appendix, we test on a simple and exactly solvable case (the *star graph* with $N$ nodes) the hypothesis that the accuracy of our approximate formula (5) may vary (within the *same* instance) from node to node, depending on how well connected the target node is to the rest of the network.

The star graph consists of a central node that is connected to all other nodes (leaves), while each leaf is only connected to the central node (has degree 1). Since the degree of the central node is therefore $N-1$, and our formula (5) is only sensitive to the incoming weights into the target, we may expect some discrepancy in how well (5) works between the the two situations (i) MFPT $m_{i1}$ between a leaf $i$ and the central node (labelled by 1), and (ii) MFPT $m_{1j}$ between the central node and a leaf $j$. Since the heterogeneity between central node and leaves increases with $N$, we expect that such discrepancy should also get larger for bigger networks.

Considering the standard (unbiased) diffusion for simplicity, the full $N \times N$ transition matrix $T$ reads

$$T = \begin{pmatrix} 0 & \frac{1}{N-1} & \frac{1}{N-1} & \cdots & \frac{1}{N-1} \\ 1 & & & & \\ 1 & & & \mathbf{0} & \\ \vdots & & & & \\ 1 & & & & \end{pmatrix} .$$

Clearly, $m_{i1} = 1$ identically for all $i \geq 2$, which – on top of being obvious given the geometry of the system – emerges from an exact evaluation of Eq. (3), as the reduced transition matrix $T^{(1)}$ is simply the $(N-1) \times (N-1)$ null matrix in this case. For this scenario, our approximate formula (5) provides the same (exact) result, as $T_{i1} = 1$, which kills the second fraction in (5).

Conversely, to compute the matrix $(\mathbb{1} - T^{(j)})^{-1}$ exactly for $j > 1$ – necessary to deal with

case (ii) – we need to use the block-matrix inversion formula that yields

$$(\mathbb{1} - T^{(j)})^{-1} = \left(\begin{array}{c|ccccc} N-1 & 1 & 1 & 1 & \cdots & 1 \\ \hline N-1 & 2 & 1 & 1 & \cdots & 1 \\ N-1 & 1 & 2 & 1 & \cdots & 1 \\ \vdots & \vdots & \vdots & \vdots & \ddots & \vdots \\ N-1 & 1 & 1 & 1 & \cdots & 2 \end{array}\right).$$

This gives $m_{1j} = 2N - 3$ for all $j > 1$. In this scenario, though, our approximate formula gives a different result, namely $m_{1j} \approx 1 + (N-1)(N-2)$. The two formulae (exact and approximate) yield the same numerical value for low $N$ ($N = 2, 3$), with the discrepancy growing with $N$. This means that our approximate formula becomes less and less accurate the fewer connections the target node has with the rest of the network. This simple example, therefore, corroborates the intuition that the accuracy of our main formula (5) can vary from node to node within the *same* instance, depending on how "well-connected" the target node is to the rest of the graph.

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
