# Peer review of ""Spectrally gapped" random walks on networks: a Mean First Passage Time formula"

_SciPost Physics, doi:SciPost Phys. 11, 088 (2021)_

## Round 2 · Referee Report · Anonymous (Referee 1) · 2021-7-20

Strengths

  1. explicit formula to compute MFPT on weighted graphs

  2. the formula requires local calculations, no matrix inversion which can be computationally cumbersome and numerically inaccurate.

  3. validity limits are theoretically established exploiting random matrix theory and replica calculations

Weaknesses

  1. the proposed formula is an approximation that only works on moderately dense graphs, there is no simple way to link the inaccuracy on sparse graphs with their local structural properties (average degree, etc.).

Report

The work by S. Bartolucci and coworkers provides an approximate but explicit formula for the Mean First Passage Time of a random walk between a source and a target node of a directed weighted network. The main idea behind this work is that of mapping the definition of MFPT onto the calculation of average quantities in a random matrix ensemble, in which a rank-1 deterministic NxN matrix (corresponding to the average weighted adjacency matrix) is perturbed by non-symmetric (and possibly correlated) gaussian random variables. In the case of i.i.d. gaussian variables, such an average can be carried out analytically using standard replica methods and a saddle-point calculation. The final formula for MFPT appears to be a sort of “noise-dressed” generalisation of known Sherman-Morrison formula for matrix inversion. It is worth stressing that, even in the large N limit, the final formula is approximate because it requires a sufficiently fast cumulant decay condition for large N. Such condition is usually verified in dense weighted networks and it can be phenomenologically related to the existence of a sufficiently large spectral gap between the isolated Perron-Frobenius eigenvalue and the bulk of the spectrum. As expected, the accuracy of the results obtained by the proposed explicit formula deteriorates increasing network sparseness. The accuracy of the approximation is tested numerically with different types of random weighted directed graphs: fully-connected networks, Erdos-Renyi random graphs and random regular graphs. In random graphs with N=500 nodes, the error of the approximate estimate compared to the exact one is just of the order of few percents for pretty high sparseness, e.g. average degree of O(10). The main advantage of this formula compared to existing approaches is that (1) it does not rely on matrix inversion, which can be cumbersome and numerically inaccurate, and (2) it is locally defined on the network. The fact that the approximate formula works well also on moderately sparse graphs (and that its limitations are pretty clearly established by a sound theoretical analysis) makes the present work an unquestionable contribution to the field of stochastic processes on random structures and complex networks.

The main result is sound and relevant, it provides a novel link between different research areas such as random matrices, disordered systems and complex networks.
The paper is well organised, clearly written, with a very pedagogical discussion of the physical meaning of the formula, which turns out to be as useful for the general public as the more technical calculations. The introduction to the problem is non-technical and accessible to a broad interdisciplinary audience. Relevant literature is cited and the state of the art is explicitly discussed in the manuscript. All theoretical results are reproducible from the calculations and numerical results could be with a straightforward algorithmic implementation of the main formula. Conclusions are clear, and in addition to summarise results, they provide some perspectives for future work. For all these reasons I think the manuscript meets all journal’s acceptance criteria and strongly recommend its publication almost in the current form (some minor amendments are suggested below).

Minor points:
— In Section 1.2, the sentence “The formula (5) is strikingly simple, not relying on a large-N approximation, …” is a bit misleading. The derivation of the formula uses a saddle-point calculation with a decay condition on some cumulant; in this sense it seems to rely on N being large (on a specific random matrix ensemble). However, I understand that the formula can be applied on any graph, for finite N, and in practice its validity only requires that the spectral gap is large enough. Is this what you mean? Please, just explain a bit better this sentence in order to avoid confusion in the reader.
— In the case of random regular graph the accuracy of the formula deteriorates considerably for low degree. In Table 2, the case for c=4 is surprisingly bad if compared with the errors reported for larger degrees and to the case of Erdos-Renyi random graphs. Do you have any idea about the reason? Could it be related to the fact that in random regular graphs entries of the adjacency matrix are much more correlated than in Erdos-Renyi graphs?
— Again on the same point, a random graph of N=500 nodes with an average degree of O(10) is not very sparse. In order to be more conclusive on the applicability of the formula on sparse graph it would be useful to show results (e.g. the scatterplots or some other metrics) at fixed average degree for different values of N (e.g. N=500,1000, 5000).
— The Authors say that “We do not report here on (dense) scale-free topologies, whose phenomenology is very similar to the other cases, albeit with significantly larger fluctuations: a detailed study of this (and other) heterogeneous cases is deferred to a separate publication.” And also “While our approach continues to work for (sufficiently dense) heterogeneous networks, the intra-row fluctuations around the random matrix assumption ⟨aij⟩ = zi/N may be very significant there and will thus require a more careful treatment“.
I do not criticise the decision of deferring the analysis of heterogenous structures to a different publication, however the present paper would really benefit of a deeper discussion on the limitations to the applicability of the formula in the case of heterogenous networks. In fact, from the fact that the formula only cares about incoming weights, one would expect that it could work pretty well when computing MFPT on hubs and central nodes and possibly much worse on poorly connected regions of the graphs. Is this correct? Is there any known result on the spectral property (spectral gap) of heterogenous graphs which prevents/limits the applicability of the formula to such graphs?

Requested changes

No major changes, some minor revisions are suggested (recommended) in the report.

  • validity: high
  • significance: high
  • originality: high
  • clarity: top
  • formatting: excellent
  • grammar: excellent

Author:  Pierpaolo Vivo  on 2021-08-18  [id 1682]

(in reply to Report 1 on 2021-07-20)
Category:
answer to question

see attached pdf

Attachment:

RebuttalLetter_FINAL_iPHui45.pdf

---

## Round 2 · Referee Report · Anonymous (Referee 2) · 2021-7-22

Report

Dear Editor

The manuscript deals with the interesting and important problem of the mean first passage time on random walks on networks. The main result, summarized in Eq. (5), is based on a previous work or the same authors (refs 36-37) and provides an elegant formula for the mean first passage time in terms of the transition matrix without any need to invert matrices, iterate them etc. In that sense, this work does provide a step forward in the field and a practical result. Furthermore, this formula is applicable to weighted and directed networks, which is an obvious strength. The main limitation of this work is the fact that it requires a significant spectral gap of the reduced transition matrix in order to perform well. In practical terms, this limits the applicability to very sparse networks. My main criticism about this manuscript in its current form is that it is not 100% well written, and could benefit from further clarifications and examples. Overall, I think that this work represents an important contribution and should be published once the authors consider the following comments:

  1. Right at the beginning (around eqs (1)-(2)) the authors define the adjacency matrix A and the transition matrix T. traditionally the adjacency matrix is a binary matrix, with 0/1 entries. I believe that the authors intend to use here a weighted version of A, and this should be stated clearly. Furthermore, it would be useful to stress that A does not have to be symmetric so that it encodes directed graphs too.

  2. In chapter 2 the authors use a random matrix A which, if I understand correctly, is not the adjacency matrix, but a different object - namely the reduced transition matrix. I find this notation confusing, and it would be best to replace the notation. It would also be useful to explain in section 2 that A is not the adjacency matrix.

  3. In Eq. (7) the authors define in terms of the z_i. It is unclear at this point how the z_i are actually defined, nor what these number actually mean. The precise definition of the z_i's is given at the last paragraph of section 2, but should appear next to Eq. (7). Also, it would be a good idea to try and motivate or provide an intuition for that choice.

  4. In the context of Eq. (13) the authors say that "it is rather straightforward to show that...| It would be useful to add a reference here, or provide an explanation.

  5. In Eq. (21) the authors formulate the cumulant decay condition. While this is a sound technical definition, it would be useful to try and provide some insight into scenarios where this is obeyed vs. scenarios where this is not obeyed.

6.In Sec. 3.2 the authors consider Erdos-Renyi networks. It is not completely clear if they use here a directed or undirected version of the network, and this should be explained. Also, I guess that the weights are incorporated into the adjacency matrix directly, and this should be stated more clearly.

  1. A second issue with ER networks is regarding the reference to the strongly connected component of the ER network. The authors correctly mention at the end of the section that c=ln N=6.21 is the connectedness threshold. I believe it would be useful to mention this background fact at the beginning of the section to remind the readers of this fact. Furthermore, in the example presented in Fig. 3 the authors use c=7 and c=190. These values are already above c=6.21 and hence the issues with the connectedness threshold are ignored. I believe that it would be useful to present a sparser ER network example as well, say for c=4. I understands that in this limit the spectral gap is much smaller and the validity of the formula is weaker, but I believe it will be useful to probe more this limit in addition to the c=7 and 190. Furthermore, the giant component of the ER network exhibits degree-degree correlations (it is disassortative) below the connectedness threshold, and it would be interesting to see the effect of this.

  2. In the context of ER as well as Random Regular Graphs, in the limit of large c's - would it be possible to provide a large c approximation for Eq. (5)? As is well known, the spectrum of such dense ER/RRG networks approaches a simple form and it might yield a simple formula for the Mean First Passage Time in this limit.

  3. In Fig. 4 the authors present results for c=7 and c=190 for Random Regular Graphs. It would be useful to include c=4 too to probe the less dense limit, where the assumptions of Eq. (5) break down. I thin this could add insight into the result. Furthermore, already for c=7 we see deviations. Can the author explain whether these deviations are systematic? It seems that the approximation tends to underestimate the exact result, hence most dots lie below the line. Is it a consistent result over many realizations or an accidental one?

  4. In the captions of Tables 1 and 2, the authors report a comparison of "numerical", exact and approximate results. First, by "numerical" they really mean results of a simulations (as written in the main text). Therefore, it would be better to replace "numerical" by simulations. Also, it would be useful to add a specific reference to the equation used for the approximate and exact results - which equations were precisely used here? Another issue – the table reports a specific pair (i,j) – which pair? What were the degrees of the nodes i and j? why not average over all pairs?

  5. On the broader picture - could the authors say anything about the distribution of first passage times? What quantity would be needed, from the random matrix side, in order to provide a similar result for the full distribution? Any comment here would be highly appreciated.

To summarize, this work provides a useful result for the mean first passage time in weighted and directed networks. I would be happy to recommend publication of the paper once the authors consider the comments mentioned above.

  • validity: top
  • significance: high
  • originality: high
  • clarity: good
  • formatting: good
  • grammar: excellent

Author:  Pierpaolo Vivo  on 2021-08-18  [id 1681]

(in reply to Report 2 on 2021-07-22)
Category:
answer to question

see attached pdf

Attachment:

RebuttalLetter_FINAL.pdf

---

## Round 3 · Referee Report · Anonymous (Referee 2) · 2021-9-3

Strengths

  1. explicit formula to compute MFPT on weighted graphs

  2. the formula requires local calculations, no matrix inversion which can be computationally cumbersome and numerically inaccurate.

  3. validity limits are theoretically established exploiting random matrix theory and replica calculations

Weaknesses

  1. the proposed formula is an approximation that only works on moderately dense graphs, there is no simple way to link the inaccuracy on sparse graphs with their local structural properties (average degree, etc.).

Report

The authors have considered satisfactorily all the comments raised by the referees and the paper should be accepted without further delay.

Requested changes

None

---

## Round 3 · Referee Report · Anonymous (Referee 1) · 2021-9-14

Strengths

  1. analytical formula for the MFPT on dense weighted graphs, obtained avoiding matrix inversion.

  2. numerical validation of the results

  3. validity limits are well established and verified numerically.

Weaknesses

  1. not so useful on sparse graphs, where the main assumptions of the calculation are not satisfied

Report

The Authors have seriously considered my previous comments and revised the manuscript accordingly. In my opinion, the paper was already well written and of certain interest for a broad audience of readers and after revision now meets the acceptance criteria on SciPost. I recommend it for publication in the current form.

Requested changes

none

---

## Round 3 · Author Response

See pdf attached as a reply to the Referees' comments.

---

## Editorial Decision

published